# Mapping QTLs for Breast Muscle Weight in an F_2_ Intercross between Native Japanese Nagoya and White Plymouth Rock Chicken Breeds

**DOI:** 10.3390/life11080754

**Published:** 2021-07-27

**Authors:** Akira Ishikawa, Bothaina H. Essa, Sherif M. Nasr, Sae Suzuki

**Affiliations:** 1Laboratory of Animal Genetics and Breeding, Graduate School of Bioagricultural Sciences, Nagoya University, Chikusa, Nagoya 464-8601, Japan; boss_bne2011@yahoo.com (B.H.E.); ss.mmz937@gmail.com (S.S.); 2Faculty of Veterinary Medicine, Damanhour University, Damanhour 22511, Egypt; shrif.menshawy@vetmed.dmu.edu.eg

**Keywords:** breast muscle weight, chickens, Nagoya, QTL, White Plymouth Rock

## Abstract

Nagoya (NAG), a native Japanese chicken breed, has high quality meat but low meat yield, whereas White Plymouth Rock (WPR), a parental breed of commercial broilers, has rapid growth but high body fat. We previously reported three quantitative trait loci (QTLs) for early postnatal growth in 239 F_2_ chickens between NAG and WPR breeds. In this study, using the same F_2_ chickens at 4 weeks of age, we performed genome-wide QTL analysis for breast muscle weight, fat weight and serum and liver levels of biochemical parameters. Two significant QTLs for pectoralis minor and/or major weights were revealed on chromosome 2 between 108 Mb and 127 Mb and chromosome 4 between 10 Mb and 68 Mb. However, no QTL for the other traits was detected. The two QTLs explained 7.0–11.1% of the phenotypic variances, and their alleles derived from WPR increased muscle weights. The chromosome 2 QTL may be a novel locus, whereas the chromosome 4 QTL coincided with a known QTL for meat quality. The findings provide information that is beneficial for genetic improvement of meat yield for the lean NAG breed and, furthermore, provide a better understanding of the genetic basis of chicken muscle development.

## 1. Introduction

More than 8000 years ago, chickens were primarily domesticated from red junglefowls in East and Southeast Asia as a main resource of food and for other reasons including religious events, ceremonial events and sports [1,2]. Since then, chickens have become major farm animals and also have been used as model animals for medical and agricultural studies [3,4,5,6].

Intensive artificial selection in broiler chickens resulted in rapid improvements in body weight, growth rates and reduction in slaughter age. Unfortunately, the selection has been accompanied by excessive subcutaneous and abdominal fat deposition. Modern broiler chickens contain 150–200 g fat per kg body weight, and approximately 85% of the fat has no economic value, and hence, it is usually disposed of during the processing of broilers [7,8]. In addition, fatness causes depression of feed efficiency, lower carcass yield, sudden death syndrome, reduction in the overall fitness, and consumer rejection for human health [9,10]. Thus, it is necessary to establish genetic improvement methods to minimize fat deposition as much as possible. The establishment of such methods will be of great value for broiler breeding programs.

In Japan, there are approximately 50 native Japanese breeds, which have mainly been developed for enjoyment of ornamental appearance, crowing and behavioral traits [11]. Little attention has so far been paid to the utility of native Japanese breeds as food resources. However, Nagoya (NAG), also known as the Nagoya Cochin, is a native Japanese breed that produces high quality meat and eggs, compared to commercial broiler and layer chickens [11]. Several native Japanese breeds have often been crossed with Western breeds to produce Japanese brand chickens that yield delicious meat [12]. These facts show the importance of native Japanese breeds as valuable genetic resources for the improvement of meat and egg quality in commercial chickens.

We previously showed marked differences in body weight, growth rate, organ weight, fat weight and breast muscle weight at 7 weeks after hatching between the NAG breed and the White Plymouth Rock (WPR) breed, which has been used worldwide as a dam breed of common broiler chickens. NAG was characterized as a lean breed with slow growth, whereas WPR was characterized as a fat breed with rapid growth [13]. Approximately 1.5 and more fold differences in those traits between NAG and WPR were observed at just 4 weeks of age [14]. NAG and WPR are reported to be genetically distinct in breed origins [15], which can provide many informative markers between the two breeds to map quantitative trait loci (QTLs) affecting the trait differences. We previously succeeded to identify three QTLs for body weights and body weight gains at 1–4 weeks of age on chromosomes 1, 2 and 4 in an F_2_ population of 239 chickens between the two breeds using 313 SNP markers obtained by restriction site-associated DNA sequencing (RAD-seq) analysis [16].

In the present study, using the same F_2_ population as that of our previous study [16], we performed genome-wide QTL analysis to identify chromosomal positions associated with breast muscle weight, fat weight and serum and liver levels of biochemical parameters. To the best of our knowledge, no QTL study on those traits has so far been performed using the NAG breed.

## 2. Materials and Methods

### 2.1. Animals and Traits

An F_2_ population of 239 chickens (119 males and 120 females) was previously obtained by an intercross between three NAG males and four WPR females [14]. Details of chicken rearing and handling were previously described [13]. At 4 weeks of age, the chickens were weighed and fasted for 2–3 h before dissection. After blood was collected from the wing vein without anticoagulant, the chickens were slaughtered under anesthesia using isoflurane. The weights of the pectoralis minor muscle and pectoralis major muscle for both left and right sides were recorded. Breast muscle weight was the sum of the pectoralis minor and major weights. The weights of abdominal fat inside the peritoneal cavity, ventriculus fat surrounding the gizzard and subcutaneous fat surrounding the neck were measured. Total fat weight was the sum of the three fat weights. In addition, percentages of the muscle and fat weights to body weight at 4 weeks of age were calculated. Serum and liver levels of triglycerides (TG), total cholesterol (TC), high density lipoprotein (HDL) and glucose were assayed as previously described [13,14]. All trait data for the F_2_ chickens and their NAG, WPR, F_1_ chickens are shown in our previous report [14].

### 2.2. Linkage Map

A linkage map of 313 SNP markers obtained by RAD-seq analysis was previously constructed using the Kosambi map function [16]. Briefly, the SNP markers were distributed in 27 linkage groups on 21 autosomes. Chromosome 1 was divided into four linkage groups, and chromosomes 4, 8 and 9 were each divided into two linkage groups (see Appendix A). The total length of the linkage map was 2125.8 cM, with an average marker spacing of 7.4 cM. The physical map positions (Mb) of the SNP markers were based on the chicken reference genome assembly Gallus-gallus-5.0 (galGal5).

### 2.3. QTL Analysis

Before QTL analysis, the effects of five environmental factors (dams, sexes, rearing persons, dissection persons and hatching dates) on phenotypic traits were tested using a linear model of the JMP Pro software version 15.2.1 (SAS Institute Japan Ltd., Tokyo, Japan), as previously described [16]. After the environmental effects that were significant at nominal 5% levels were removed from the raw trait data, the residuals were tested for significant deviation from normality by the Shapiro-Wilk test of JMP Pro. The residuals that did not meet normality at the nominal 5% level were approximated to normality by Johnson’s family transformation or the other methods. The transformed data were standardized to SD unit to allow easy comparisons of the QTL effects among the traits.

QTL analysis was performed using the Haley-Knott regression by the function calc.genoprob of the R/qtl package version 3.5.1 [17]. To find QTLs with main effects on traits, a single-QTL genome scan with a single QTL model was carried out for sex-combined data by the function scanone of R/qtl. Logarithms of the odds (LOD) scores were computed at a 1-cM interval across the linkage map with a genotyping error rate (error.prob) of 0.01. To map QTLs with additive effects and/or epistatic interaction effects, a two-dimensional genome scan with a two-QTL model was performed by the function scantwo of R/qtl. LOD scores were calculated at a 5-cM interval across the linkage map with error.prob = 0.01. Genome-wide significance threshold levels at 5% significant levels and 10% suggestive levels were calculated with 10,000 permutations for scanone and 500 permutations for scantwo.

To detect QTLs with sex-specific effects on traits, a single-QTL genome scan was performed using sex-combined data by including sex as additive and/or interactive covariates in a single QTL model. As previously described [17], LOD scores with sex as additive and interactive covariates in the model were calculated and LOD scores with sex as only an additive covariate in the model were calculated. Differences between the two LOD scores calculated were LOD scores for QTL-by-sex interaction effects. Likewise, genome-wide significance thresholds for the QTL-by-sex interactions were calculated by subtracting the LOD thresholds obtained by 10,000 permutations with sex as an additive covariate from the LOD thresholds obtained by 10,000 permutations with sex as additive and interactive covariates. If the LOD scores for the QTL-by-sex interactions exceeded the genome-wide significance threshold levels, then a single-QTL genome scan and permutation tests were performed for each sex separately, as described above. When suggestive or significant QTLs were found in one sex at the same map positions as those of the QTL-by-sex interactions, the presence of sex-specific QTLs was declared.

For each QTL detected, a 1.8-LOD drop (comparable to 95%) confidence interval (CI), a percentage of phenotypic variance explained (% Var), and additive and dominant effects were computed by the function fitqtl of R/qtl [17]. The additive effect was half of the trait difference between two homozygotes for the allele derived from the NAG breed and the allele derived from the WPR breed. The dominant effect was the difference between heterozygotes for NAG and WPR alleles and the average between two homozygotes. The trait differences among three genotypes at the SNP marker nearest to the QTL detected were tested by one-way analysis of variance (ANOVA) followed by Tukey’s honestly significant difference (HSD) post hoc test of JMP Pro.

## 3. Results

### 3.1. QTL Analysis for Breast Muscle Weight

As shown in Figure 1 and Appendix A, single-QTL genome scans revealed at least two QTLs associated for three weights of the pectoralis minor muscle, pectoralis major muscle and breast muscle (the sum of minor and major muscles) on chromosome 2 and linkage group 4b assigned to chromosome 4. On chromosome 2, single peaks of LOD scores of 3.8–6.0 that exceeded 5% significance threshold levels were observed for the three muscle weights at nearly the same map position of 83–84 cM (116.4–116.8 Mb). In contrast, on linkage group 4b, two LOD peaks exceeding 10% or 5% significance threshold levels were observed for pectoralis minor weight (Figure 1), suggesting the possibility of the presence of two QTLs. However, a two-dimensional genome scan for this trait showed no statistical evidence for the presence of two QTLs at genome-wide 10% or less threshold levels. Likewise, no such evidence was shown for the other muscle weights on linkage group 4b. Furthermore, neither epistatic QTL nor sex-specific QTL was identified for the three muscle weight traits.

As summarized in Table 1, the chromosome 2 QTL for three weights of the pectoralis minor, pectoralis major and breast muscle explained 11.1, 7.1 and 8.0% of the phenotypic variances, respectively. The QTL was located in a 95% confidence interval of 49–101 cM (108.1–126.6 Mb). The chromosome 4 QTL for the three muscle weights explained 6.6–7.0% of the phenotypic variances. The QTL was located in a 95% confidence interval of 13–137 cM (10.4–67.7 Mb). At both QTLs, the allele derived from WPR increased muscle weights (Table 1 and Figure 2). However, the mode of inheritance was different between the two QTLs. The mode of the WPR allele was autosomal recessive at the chromosome 2 QTL, whereas it was autosomal additive at the chromosome 4 QTL.

Single-QTL genome scans for the percentages of the three muscle weights to body weight at 4 weeks of age revealed a QTL with a peak LOD score of 5.1 for only the percentage of the pectoralis minor weight at 84 cM (116.4 Mb) on chromosome 2 at the genome-wide 5% significance level (Appendix A). As summarized in Table 1, this QTL explained 9.5% of the phenotypic variance, and it was located in a 95% confidence interval of 61–102 cM (107.6–123.6 Mb). The WPR allele at the QTL was inherited in an autosomal recessive manner (Appendix A). Neither epistatic nor sex-specific QTL was detected for the percentages of three muscle weights. 

Taken together, the results indicated that two QTLs for breast muscle weight and/or pectoralis minor percentage were located on chromosomes 2 and 4 with a different mode of inheritance between the two QTLs. The phenotypic effect of the chromosome 2 QTL was greater than that of the chromosome 4 QTL. Importantly, the chromosome 2 QTL exerted a greater effect on pectoralis minor weight and its effect was independent from body weight.

### 3.2. QTL Analysis for Fat Weight

Single-QTL genome scans revealed no QTL for weights of subcutaneous, abdominal, ventriculus and their total fat at genome-wide 10% or less levels (Appendix A). Likewise, no QTL for the percentages of those fat weights to body weight at 4 weeks of age was identified (Appendix A). In addition, two-dimensional genome scans revealed neither epistatic QTL nor sex-specific QTL for any fat traits at genome-wide 10% or less levels.

### 3.3. QTL Analyses for Serum and Liver Levels of Biochemical Parameters

Single-QTL genome scans revealed no QTL for serum and liver levels of TC, TG, HDL and glucose at genome-wide 10% or less levels (Appendix A). In addition, two-dimensional genome scans revealed neither epistatic QTL nor sex-specific QTL for any biochemical parameters at genome-wide 10% or less levels.

## 4. Discussion

This study revealed two QTL regions for pectoralis minor and major weights on chromosomes 2 and 4. The chromosome 2 region showed a single LOD peak for both minor and major weights, indicating the presence of a single QTL. However, the chromosome 4 region showed two LOD peaks for pectoralis minor weight. The first proximal peak exceeded the genome-wide 5% level. The second distal peak exceeded the genome-wide 10% level, but it did not show statistical evidence for an additional locus for pectoralis minor weight. By contrast, the second peak showed evidence for a suggestive QTL for pectoralis major weight. In mice, multiple closely linked QTLs for growth and body composition traits were revealed on a small chromosomal region of 44 Mb [18]. In that region, two tightly linked QTLs with opposite effects on body weight were discovered by fine mapping using congenic and subcongenic strains of mice [19]. Thus, it is very likely that the QTL region of chicken chromosome 4 contains two or more numbers of loci for breast muscle weight. However, it will be difficult to resolve the problem of closely linked loci without further fine mapping of the QTL region.

Many QTLs for breast muscle weights and breast muscle percentages have previously been reported on chicken chromosomes 2 and 4, and most of the QTLs have been deposited in the Chicken Quantitative Trait Locus Database, chickenQTLdb [20]. Some QTLs deposited in chickenQTLdb have broad confidence intervals and their map positions are not physically defined due to the use of linkage maps (cM) based on microsatellite markers and other reasons. Hence, we compared our QTLs to the previous QTLs with the physical map positions (based on galGal5 assembly), which were determined by using SNP markers or approximated by chickenQTLdb.

On chromosome 2 between 107.46 Mb and 107.91 Mb, a QTL for pectoralis minor weight was previously reported in an F_2_ intercross between fat and lean broiler lines divergently selected for abdominal fatness [21]. That interval did not overlap with our QTL interval between 108.1 Mb and 126.6 Mb. A QTL for dry matter in breast muscle was reported at 76 Mb by a genome-wide association study (GWAS) in an F_2_ population between a Chinese local breed and a commercial broiler line [22], which is distant from our QTL. The interval of our QTL did not overlap with those of any other QTLs previously reported for breast muscle weights and breast muscle percentages. It is, thus, suggested that our chromosome 2 QTL may be a novel locus. However, we cannot rule out the possibility that previous QTLs with physically undefined map positions are coincident with our locus.

On chromosomes 4, a QTL for pectoralis major yield and breast muscle yield at 6 weeks of age was previously revealed at 65.97 Mb by GWAS using two broiler lines divergently selected for pectoralis major ultimate pH, which reflects the level of muscle glycogen content, a determining criterion of meat quality [23]. This map position is very close to that of the LOD peak for our chromosome 4 QTL at approximately 65 Mb. It is generally known that NAG has superior meat quality to commercial broiler chickens including WPR [11]. Hence, it is likely that our QTL may have a pleiotropic effect on muscle glycogen content. A QTL for breast muscle mass in an advance intercross lines between New Hampshire and White Leghorn was finely mapped to a region of 74.5–78.0 Mb [24], which is located outside of our QTL interval between 10.4 Mb and 67.7 Mb. No other QTL previously reported for breast muscle weights and breast muscle percentages coincided with our QTL.

A previous QTL study on breast muscle weight used WPR and Oh-Shamo as two parental breeds to develop an F_2_ mapping population, and it revealed a significant QTL for the percentage of the pectoralis minor weight to carcass weight on chromosome 24 and two suggestive QTLs for pectoralis minor and major weights on chromosomes 3 and Z [25]. Interestingly, the strain of the WPR breed used in the previous study was the same as that of our WPR. However, the QTL positions revealed are completely different between the previous study and our study. Oh-Shamo, another native Japanese breed, is reported to be greatly different in genetic relationship based on microsatellite markers and breed origin from NAG used in our intercross [11,15]. Therefore, the difference in QTL positions between the two studies is considered to reflect the genetic difference between Oh-Shamo and NAG breeds.

Since the confidence intervals of our two QTLs still contain hundreds of genes, it is premature to search for positional candidate genes with known or unknown functions without further fine mapping of our loci. However, a previous study has reported the platelet-derived growth factor receptor alpha (*PDGFRA*) gene and the sarcoglycan beta (*SGCB*) gene as candidate genes for a QTL for pectoralis major ultimate pH on chromosome 4 [23]. *PDGFRA* is located at 65.8 Mb and is known to be a marker of fibro-adipogenic precursors, which have been characterized by playing a role in muscle regeneration and repair [26]. *SGCB* is located at 66.6 Mb and encodes the beta sarcoglycan protein, which is a member of the sarcoglycan complex located to the sarcolemma. Mutations in this gene were associated with human limb-girdle muscular dystrophy [27]. Beta-sarcoglycan-deficient mice showed progressive muscular dystrophy with extensive degeneration and regeneration of muscle fibers; furthermore, the mice exhibited muscular hypertrophy and whitish stripes in the muscles [28]. Therefore, the two genes can become positional and functional candidate genes for our chromosome 4 QTL because our locus is located near the previous QTL, as described earlier.

## 5. Conclusions

This study revealed two QTLs affecting breast muscle weight and pectoralis minor percentage at 4 weeks of age on chicken chromosomes 2 and 4 in the F_2_ intercross population between NAG and WPR breeds. The chromosome 2 QTL may be a novel locus, whereas the chromosome 4 QTL coincided with a known locus for meat quality. These findings provide information that is beneficial for genetic improvement of meat yield for NAG with lower muscle weight than WPR. In addition, the study was the first step to identification of causal quantitative trait genes (QTGs) for the difference in breast muscle weight between the two breeds. The identification of the QTGs will provide new insights into the genetic basis of chicken muscle development.

## Figures and Tables

**Figure 1 life-11-00754-f001:**
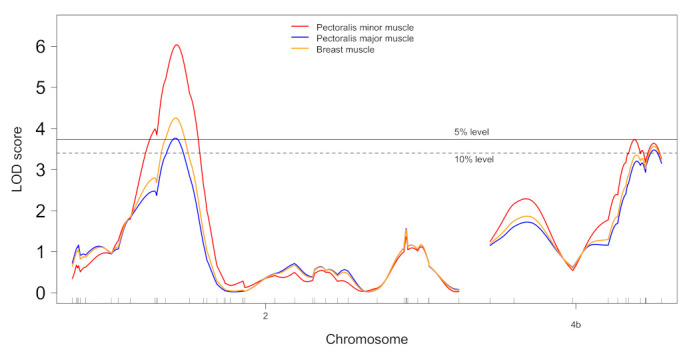
LOD score plots for QTLs affecting weights of the pectoralis minor muscle, pectoralis major muscle and breast muscle on chromosome 2 and linkage group 4b assigned to chromosome 4. Horizontal dashed and solid lines show genome-wide 10% and 5% significance threshold levels (see Appendix A for details).

**Figure 2 life-11-00754-f002:**
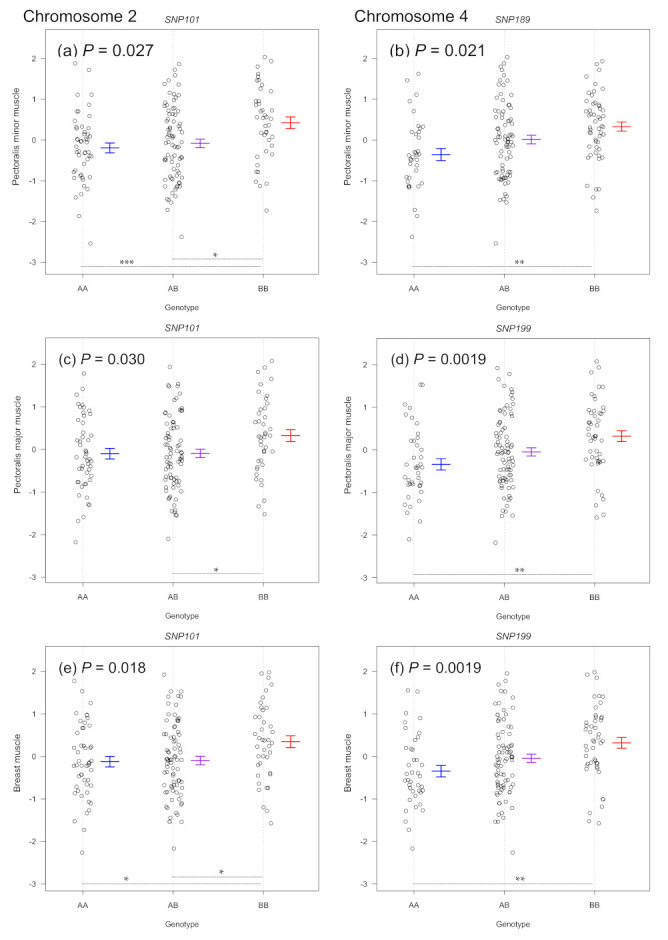
Genotype effect plots of the SNP markers nearest to QTLs for weights of (**a**,**b**) pectoralis minor muscle, (**c**,**d**) pectoralis major muscle and (**e**,**f**) breast muscle on chromosomes 2 and 4. Each circle shows muscle weight in SD unit for an animal. Horizontal solid bars for each genotype show mean ± SE. *P* values were obtained from a one-way ANOVA. * *P* < 0.05, ** *P* < 0.01 and *** *P* < 0.001 on horizontal dashed lines refer to statistical differences between two genotypes (Tukey’s honestly significant difference test). A: NAG allele, B: WPR allele.

**Table 1 life-11-00754-t001:** Parameter estimates of QTLs for breast muscle weights and percentages in the F_2_ cross between NAG and WPR breeds.

Trait	Chr. ^1^	Position ^2^	LOD	Nearest Marker	CI ^3^	Additive Effect ^4^	Dominant Effect ^4^	% Var
Pectoralis minor muscle	2	84 (116.4)	6.0 **	*SNP101*	69–99 (109.2–120.6)	0.53 ± 0.10	−0.25 ± 0.18	11.1
	4b	116 (65.1)	3.7 *	*SNP189*	13–137 (10.4–67.7)	0.39 ± 0.10	0.18 ± 0.15	7.0
Pectoralis major muscle	2	83 (116.8)	3.8 *	*SNP101*	49–101 (108.1–126.6)	0.42 ± 0.10	−0.25 ± 0.18	7.1
	4b	131 (65.9)	3.5	*SNP199*	23–137 (48.4–67.7)	0.40 ± 0.10	0.04 ± 0.15	6.6
Breast muscle	2	83 (116.8)	4.3 *	*SNP101*	56–100 (108.7–125.5)	0.44 ± 0.10	−0.25 ± 0.18	8.0
	4b	131 (65.9)	3.6	*SNP199*	23–137 (48.4–67.7)	0.40 ± 0.10	0.04 ± 0.15	6.7
% Pectoralis minor muscle	2	84 (116.4)	5.1 **	*SNP101*	61–102 (107.6–123.6)	0.50 ± 0.10	−0.20 ± 0.18	9.5

^1^ Chromosome (see Appendix A for the linkage map); ^2^ Linkage position in cM and the physical position in Mb in parenthesis; ^3^ Linkage interval in cM and the physical interval in Mb in parenthesis; ^4^ The positive sign of the additive and dominant effects (mean ± SE) in SD unit indicates that the WPR-derived allele increased the trait value; ** and * Significant at genome-wide 1% and 5% levels, respectively, while no asterisk shows genome-wide 10% significance levels (see Appendix A for significance threshold levels).

## Data Availability

All QTL results generated during the current study are shown in the Appendix A. The other datasets analyzed during the current study are available from the corresponding author on reasonable request.

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
