# Peer review of "Mapping QTLs for Breast Muscle Weight in an F2 Intercross between Native Japanese Nagoya and White Plymouth Rock Chicken Breeds"

_life, 2021, doi:10.3390/life11080754_

Round 1

Reviewer 1 Report

Dear authors,

I have two major concerns regarding this study. Firstly, no new genotypic and sample data are presented in this analysis. The animals and SNP markers used are the same as in your previous published study. The only new data are the addition of new phenotypic characters. Secondly, I have doubts regarding the statistical confidence of the results, as 239 individuals and 313 SNP markers were used. In my opinion, these numbers are a bit low in order to obtain a reliable QTL analysis... 

Author Response

Author's Reply to the Review Report (Reviewer 1):

I have two major concerns regarding this study. Firstly, no new genotypic and sample data are presented in this analysis. The animals and SNP markers used are the same as in your previous published study. The only new data are the addition of new phenotypic characters.

Reply: As Reviewer 1 pointed out and we mentioned in Introduction and Materials and Methods, this study used the same animals and sample data as in our previous study published in Anim. Genet. 2021. However, the QTLs detected in the present study have not yet been reported elsewhere. Hence, the detected QTLs are new data. In other words, the present study can be considered as a sister paper of the previous published study, in which QTLs for body weight and growth were reported. As reported in our previous study published in Damanhur J. Vet. Sci. 2019, we measured many traits for body weight, body compositions, muscle weights and blood levels. We considered that, if the results of QTL analyses for all traits were published in one paper, readers could not easily understand all QTL results due to messy data for many traits and results. So far, this kind of sister paper has been published in many fields of journals.

Secondly, I have doubts regarding the statistical confidence of the results, as 239 individuals and 313 SNP markers were used. In my opinion, these numbers are a bit low in order to obtain a reliable QTL analysis...

Reply: It is well known that there are two common methods for QTL mapping. One is a genome-wide association study (GWAS) based on linkage disequilibrium (LD) in outbred populations such as humans and large livestock animals, and the other is so-called genome-wide QTL analysis based on linkage analysis in three-generation pedigrees or designed crosses of model animals and small livestock animals such as chickens and pigs (Nat. Educ. 2008, 1, 208). Genome-wide QTL analysis should not be confused with GWAS. Even if SNP markers were used, it would not always mean that GWAS was used for QTL mapping. In the present study, we performed genome-wide QTL analysis based on linkage analysis using SNP markers. The statistical power of the genome-wide QTL analysis has already well demonstrated, and it is shown that at least 200 animals are enough to map major QTLs (e.g., Nat. Genet. 18:19, 1998). Furthermore, as we described in Conclusion, the present study is the first step to identifying causal genes for major breast muscle weight QTLs detected. And see our second reply to Reviewer 2.

Reviewer 2 Report

This is a well-written manuscript; however, a couple of concerns are evident because several sentences are copied from their previous publication. Overall, much of the manuscript is either repeating or explaining statements of the previous articles of the authors. In addition, marker density is very limited resulting in constrained power of the analyses. Given that body weight is a complex trait across all species, multiple QTLs are expected.

A few minor edits are also suggested.

Line 20: %%?

Chromosomes 1a, 1b, 1c and 1d, labels (similarly other a/b for 4, 8 and 9) on Figures and description in text are misleading. These are linkage groups on a chromosome rather than independent chromosomes.

Lines 154-161: Table 1 footnote is too long and most of the information should be part of Materials and Methods.

Author Response

Author's Reply to the Review Report (Reviewer 2):

This is a well-written manuscript; however, a couple of concerns are evident because several sentences are copied from their previous publication. Overall, much of the manuscript is either repeating or explaining statements of the previous articles of the authors.

Reply: As pointed out in the comment, some sentences in Introduction are near to copied ones. So, we rephrased such sentences as much as possible. However, we consider that, for example, the general explanation for the native Japanese breeds including NAG are needed for many general readers who are not familiar with the native Japanese chickens.

In addition, marker density is very limited resulting in constrained power of theanalyses. Given that body weight is a complex trait across all species, multiple QTLs are expected.

Reply: As described in our reply to Reviewer 1, the present study performed genome-wide QTL analysis based on linkage analysis in three-generation pedigrees. It has generally been known that a marker spacing of approximately 20 cM provides adequate power for QTL mapping in an F2 population with 250 individuals (e.g., Genomics 21:626, 1994; Nat. Genet. 18:19, 1998; Ref. 17 of the manuscript). Since the resolution of a QTL map position is dependent on recombination frequencies between markers used, it has already been demonstrated that a narrower marker spacing than 20 cM do not result in higher resolution of the QTL position because additional recombination is not so expected to occur on one chromosome in a given sample size (e.g., Genomics 21:626, 1994; Nat. Genet. 18:19, 1998; Ref. 17 of the manuscript). In the present study, we used 239 F2 animals and 313 markers with an average marker spacing of 7.4 cM. Therefore, our QTL mapping design is adequate. Since breast muscle weight, not body weight as Reviewer 2 mentioned, is a quantitative trait, we performed two-dimensional genome scan the present study, which provided no evidence of linked QTLs on a linkage group.

A few minor edits are also suggested.

Line 20: %%?

Reply: We deleted one of %. Thank you.

Chromosomes 1a, 1b, 1c and 1d, labels (similarly other a/b for 4, 8 and 9) on Figures and description in text are misleading. These are linkage groups on a chromosome rather than independent chromosomes.

Reply: Thank you for the comment. To avoid the misleading, we added the linkage map for the SNP markers as a new supplementary figure, Figure S1, and we corrected the description in the text. In our previous paper published in Anim. Genet., the linkage map was provided as a supplementary table. So, Figure S1 is not reused.

Lines 154-161: Table 1 footnote is too long and most of the information should be part of Materials and Methods.

Reply: From the footnote of Table 1 and Figure 1, we deleted or rephrased some sentences which are shown in Materials and Methods.

Reviewer 3 Report

Dear authors,

This study analyzes the presence of QTLs associated with pectoralis minor and/or major weights in Nagoya, a native Japanese chicken breed. The results obtained in this work are very interesting from the point of view of genetic improvement in chickens and for animal genetics in general. The manuscript is well structured and written, the introduction provides sufficient background and includes all relevant references, the research design is appropriate, the results are clearly presented, and the conclusions are supported by these results. However, minor changes are required. Concretely, the authors should divide the “Material and Methods” section in subsections, such as “Animals”, “Genetic análisis”, “statistical análisis”, etc. Besides, the authors should avoid subjective words in the discusion, such as “surprisingly” (line 212).

I hope that these recommendations improve the quality of the manuscript, which presents very interesenting results.

Author Response

Author's Reply to the Review Report (Reviewer 3):

This study analyzes the presence of QTLs associated with pectoralis minor and/or major weights in Nagoya, a native Japanese chicken breed. The results obtained in this work are very interesting from the point of view of genetic improvement in chickens and for animal genetics in general. The manuscript is well structured and written, the introduction provides sufficient background and includes all relevant references, the research design is appropriate, the results are clearly presented, and the conclusions are supported by these results. However, minor changes are required. Concretely, the authors should divide the “Material and Methods” section in subsections, such as “Animals”, “Genetic análisis”, “statistical análisis”, etc.

Reply: Thank you for the constructive comment. We added such subsections in Materials and Methods.

Besides, the authors should avoid subjective words in the discusion, such as “surprisingly” (line 212).

Reply: We deleted the word “surprisingly”.

Round 2

Reviewer 1 Report

Dear authors, I have already sent my comments to the Editors. I still believe that the paper in its present format should not be considered for publication in this high impact journal, as a low amount of new data are added compared to your previous publication.